# VLM-R$^3$: Region Recognition, Reasoning, and Refinement for Enhanced Multimodal Chain-of-Thought

**Chaoya Jiang**[1]*, **Yongrui Heng**[1]*, **Wei Ye**[1]†, **Han Yang**[3], **Haiyang Xu**[2]†,
**Ming Yan**[2], **Ji Zhang**[2], **Fei Huang**[2], **Shikun Zhang**[1]

[1] National Engineering Research Center for Software Engineering, Peking University
[2] Alibaba Group
[3] ZEEKR Intelligent Technology Holding Limited
{wye}@pku.edu.cn,
{shuofeng.xhy}@alibaba-inc.com

## Abstract

Recently, reasoning-based MLLMs have achieved a degree of success in generating long-form textual reasoning chains. However, they still struggle with complex tasks that necessitate dynamic and iterative focusing on and revisiting of visual regions to achieve precise grounding of textual reasoning in visual evidence. We introduce **VLM-R$^3$** (**V**isual **L**anguage **M**odel with **R**egion **R**ecognition and **R**easoning), a framework that equips an MLLM with the ability to (i) decide *when* additional visual evidence is needed, (ii) determine *where* to ground within the image, and (iii) seamlessly weave the relevant sub-image content back into an interleaved chain-of-thought. The core of our method is **Region-Conditioned Reinforcement Policy Optimization (R-GRPO)**, a training paradigm that rewards the model for selecting informative regions, formulating appropriate transformations (e.g. crop, zoom), and integrating the resulting visual context into subsequent reasoning steps. To bootstrap this policy, we compile a modest but carefully curated Visuo-Lingual Interleaved Rationale (VLIR) corpus that provides step-level supervision on region selection and textual justification. Extensive experiments on MathVista, ScienceQA, and other benchmarks show that VLM-R$^3$ sets a new state of the art in zero-shot and few-shot settings, with the largest gains appearing on questions demanding subtle spatial reasoning or fine-grained visual cue extraction.

## 1 Introduction

Multimodal Large Language Models (MLLMs) have recently emerged as a powerful paradigm, demonstrating remarkable capabilities in understanding and generating content across different modalities, primarily vision and language [38, 27, 23, 65, 5, 7]. Models like O1 [37], QvQ [3], and Gemini 2.5 [1] have showcased impressive performance on a wide array of tasks such as MMMU [66], MathVista [30], and ScienceQA [31]. A key factor contributing to their advanced reasoning abilities is the integration of Chain-of-Thought (CoT) prompting [58], which elicits step-by-step reasoning pathways, often leading to more accurate and interpretable outputs.

Despite these advancements, a critical limitation persists in the way current MLLMs interact with visual information during complex reasoning processes. Most existing approaches [3, 37, 61, 62]

---

*These authors contributed equally to this work.
†corresponding authors.

employing CoT predominantly confine the reasoning steps to the textual domain, with only an initial static grounding in the visual input. This paradigm falls short in scenarios demanding dynamic, iterative, and fine-grained interaction with specific visual regions throughout the reasoning chain.

As shown in Figure 1, examples include sequentially verifying hypotheses against image details, tracking object states across visual cues, or comprehending intricate spatial relationshipsall of which require a more active and adaptive visual grounding mechanism. Encouragingly, recent models such as O3 [2] which capable of interleaving image analysis with text generationinspire a new frontier where reasoning is not merely conditioned on an image, but is continuously intertwined with ongoing visual perception and localization.

Developing an MLLM that can look again during reasoning faces two notable hurdles: **Region-grounding learning.** The model must learn *where* to focus and *how* to transform the grounded region (crop, zoom) based on partial textual deliberation. **Credit assignment.** Simply supervising final answers does not teach the model whether a chosen region actually contributed to correct reasoning, making it hard to refine the visual-query policy.

To bridge this crucial gap, we make two primary contributions. First, we introduce Visuo-Lingual Interleaved Rationale (VLIR), a pioneering dataset meticulously curated to support the development of MLLMs for interleaved text-image CoT reasoning. VLIR provides explicit annotations for visual region localization, image cropping instructions, and semantic enhancement cues, all embedded within multi-step reasoning narratives. Second, building upon this, we propose **VLM-R$^3$** (**V**isual **L**anguage **M**odel with **R**egion **R**ecognition and **R**easoning), a novel framework designed to master this intricate reasoning style. VLM-R$^3$ is trained using a distinctive strategy that combines cold-start finetuning on our VLIR dataset with a novel Region-Conditioned Reinforcement Policy Optimization (R-GRPO). This empowers VLM-R$^3$ to

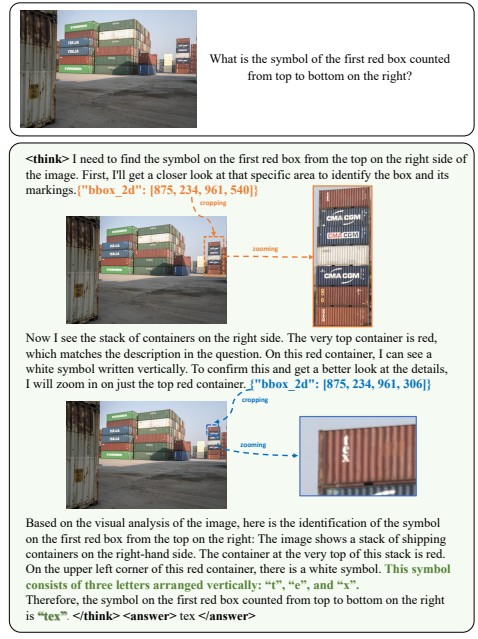

Figure 1: This figure visualizes our proposed VLM-R$^3$ approach, which integrates region grounding and refinement in an interleaved visual-textual reasoning chain. While conventional text-based reasoning fails when analyzing scenes that require dynamic, iterative, and fine-grained interaction with specific visual regions, our approach succeeds by precisely identifying and focusing on critical visual elements, such as 'the first red box counted from top to bottom on the right' in this example, to derive accurate conclusions through targeted visual reasoning.

learn when and where to look within an image, how to process the localized visual evidence (e.g., by cropping or requesting enhancement), and how to integrate this dynamically acquired information into its evolving reasoning chain. Our extensive experiments on diverse multimodal reasoning benchmarks, including MME [14], ScienceQA [31] and MathVista [30], demonstrate that VLM-R$^3$ significantly outperforms existing state-of-the-art models. In summary, our contributions are:

- The introduction of VLIR, the first benchmark dataset tailored for training and evaluating MLLMs on interleaved visual-textual CoT reasoning with explicit region-level interactions.

- The proposal of VLM-R$^3$, a novel MLLM framework, and its associated R-GRPO training strategy, which enables dynamic visual region localization and evidence integration within the reasoning process.

- Comprehensive empirical validation showing that VLM-R$^3$ achieves superior performance on challenging multimodal reasoning tasks, setting a new benchmark for fine-grained, visually-grounded inference.

## 2 Related Work

### 2.1 Large Language Model Reasoning

Reasoning in Large Language Models [50, 70, 67, 18] evolved substantially with Chain of Thought (CoT) prompting [49, 58, 24, 57, 35, 41], which enables models to break down complex problems into intermediate steps, mimicking human reasoning. This foundational approach has expanded to include diverse structures like program-of-thoughts [10], table-of-thoughts [19], and tree-of-thoughts [64], each offering unique advantages for different reasoning scenarios. Recent advances include OpenAI's O1 [37], which combines reinforcement learning [39, 44, 16] with CoT to optimize decision-making without external guidance, and DeepSeek R1 [12], which employs pure reinforcement learning through Group Relative Policy Optimization (GRPO) [46] to enable autonomous evolution of reasoning capabilities while incorporating rule-based rewards that significantly improve performance across complex reasoning tasks.

### 2.2 Multi-modal Large Language Model Reasoning

Multi-modal Large Language Model reasoning research [69, 59, 43, 34, 40, 29, 28] has emerged following the success of text-only reasoning models [37, 6, 12, 52], focusing on both effective multi-modal chain-of-thought structures [61, 53, 51, 21] and high-quality training data construction methods [13, 47, 4]. Mainstream approaches have adapted text-based reasoning paradigms to multi-modal contexts, as seen in Virgo [13], which demonstrated that text-only reasoning data can activate certain multi-modal reasoning capabilities, and more structured frameworks like LLaVA-CoT's [61] four-stage reasoning process and MM-Verify's [51] verification-enhanced approach. However, these methods largely inherit reasoning paradigms from text-only models without adequately addressing visual information processing, leading to limitations in visually-intensive reasoning tasks.

## 3 Method

We propose a novel framework, VLM-R$^3$, designed to perform visuo-lingual interleaved reasoning with region grounding. This section details the components of our approach, including the construction of the Visuo-Lingual Interleaved Rationale (VLIR) dataset used for cold-start supervised fine-tuning, the interactive inference pipeline enabling dynamic visual grounding, and the Region-Conditioned Reinforcement Policy Optimization (R-GRPO) strategy employed to enhance reasoning capabilities.

### 3.1 Visuo-Lingual Interleaved Rationale (VLIR) Dataset

Prior work, such as Visual CoT [45], introduced the concept of incorporating visual grounding (specifically bounding boxes) into reasoning chains. However, these methods typically suffer from several limitations: (1) They often lack explicit linguistic reasoning steps interleaved with visual actions. (2) The visual grounding actions (e.g., cropping based on bounding boxes) are predefined or manually specified, rather than being dynamically generated by the model. (3) They are often restricted to a limited number of visual interactions, typically a single bounding box selection before providing a final answer, lacking the flexibility for multi-step visual querying. To address these limitations and cultivate the ability for models to autonomously and flexibly perform iterative visual retrieval and cropping based on their ongoing reasoning, we introduce the **Visuo-Lingual Interleaved Rationale (VLIR)** dataset. This dataset is specifically curated to provide rich, interleaved sequences of textual reasoning steps interspersed with explicit visual grounding actions and the corresponding cropped visual evidence.

#### 3.1.1 Data Construction

The construction of the VLIR dataset focuses on scenarios that necessitate fine-grained spatial understanding and precise utilization of visual cues. We select data from a diverse set of existing benchmarks to cover a wide range of visual reasoning challenges:

- **Text/Document Understanding:** TextVQA [48], DocVQA [33] for tasks requiring OCR and document structure understanding.

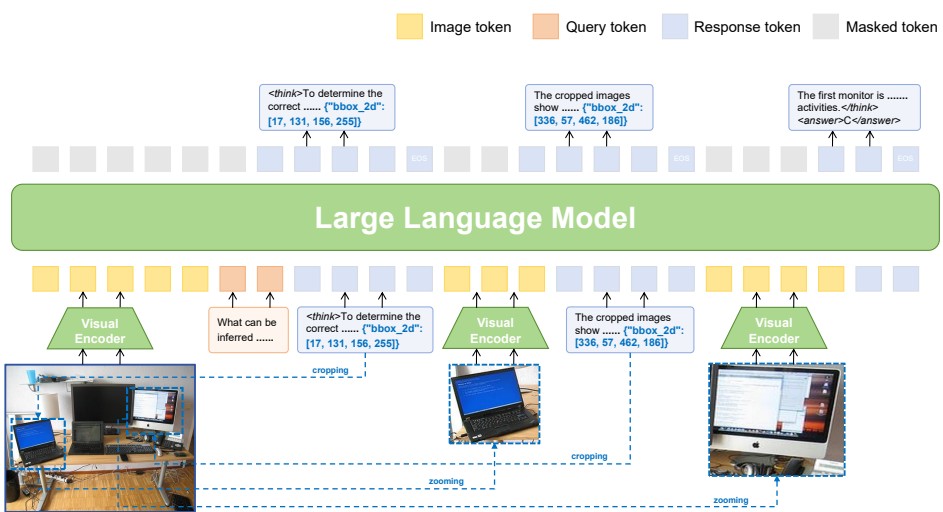

Figure 2: The visualization of the inference pipeline of our proposed methods.

- **General Visual Question Answering:** GQA [17] for complex multistep reasoning over visual scenes.

- **Chart and Infographic Interpretation:** InfographicsVQA [32] for understanding structured visual data.

- **Spatial Relation Reasoning:** VSR [26] for tasks focused on identifying and reasoning about spatial relationships between objects.

We leverage the advanced capabilities of powerful MLLMs, such as Qwen2.5-VL 72B [8], through sophisticated prompt engineering to generate interleaved image-text reasoning chains for data points from benchmarks like GQA and TextVQA, which represent real-world question answering scenarios. We then employ a rejection sampling strategy on the generated samples, filtering for those that align with the ground-truth answers.

For tasks where direct prompt engineering on the original image-question pair is less effective, particularly those involving detailed OCR or tabular data interpretation (e.g., data underlying Visual CoT [45]), we utilize GPT-4o [36] with tailored prompts that incorporate the metadata provided by the source dataset (e.g., the initial bounding boxes from Visual CoT). This allows us to generate detailed, step-by-step interleaved rationales within these challenging domains.

### 3.1.2 Data Filtering

To ensure the quality and relevance of the interleaved rationales generated, we apply a rigorous filtering process based on the following criteria:

- **Semantic Unit Validity of Regions:** Each proposed bounding box must enclose a complete and semantically meaningful visual unit (e.g., a recognizable object, a block of text, or a distinct part of a chart). To automate this, we utilize a smaller VLM and prompt it with the cropped image corresponding to the proposed bbox, asking it to confirm the presence and identity of a recognizable entity ("Can you identify what is in this image (a specific object or piece of text)? Respond with yes/no."). Samples in which the VLM fails to confirm a meaningful semantic unit are rejected.

- **Logical Coherence and Non-Redundancy of Reasoning:** The generated textual reasoning steps must be logically sound, progressive, and directly contribute to arriving at the final answer, avoiding spurious or redundant text. We employ a powerful text-only LLM, such as DeepSeek V3 [25], through prompt engineering to evaluate the logical flow and relevance of the text rationale preceding each visual interaction and the overall reasoning path. Samples with illogical or padded reasoning are rejected.

## 3.2 Interactive Inference Pipeline

The VLM-R$^3$ model executes reasoning through an interactive pipeline that enables the model to dynamically select and incorporate visual information during its inference process.

The interaction is initiated by providing the VLM-R$^3$ with an instruction that defines the reasoning task and the available visual interaction tool:

```
You need to first think about the reasoning process in your mind, and then
provide the answer. When thinking, you should call the "crop" tool (format:
{"bbox_2d": [x1, y1, x2, y2]}) to focus on the key areas in the image. The
reasoning process and the answer are included in the <think> </think> and
<answer> </answer> tags respectively.
```

When the model generates a string that matches the specified JSON format, the pipeline intercepts the output. The system parses the coordinates $[x1, y1, x2, y2]$ and performs a cropping operation on the original input image. The resulting cropped image is then zoomed in and encoded into visual tokens and appended to the model's input sequence, effectively providing the model with the requested visual detail as a new context. Following the injection of the cropped image, the model resumes generation, which may involve generating further text or issuing additional "Crop" commands. This interactive loop continues until the model generates the final answer, at which point the process terminates. This pipeline structure shows as Figure 2,

## 3.3 Region-Conditioned Reinforcement Policy Optimization (R-GRPO)

Standard supervised learning on fixed trajectories struggles to optimize the complex state-dependent policy of deciding *when* and *where* to acquire visual information. Our approach, Region-Conditioned Reinforcement Policy Optimization (R-GRPO), adapts a policy optimization framework, building upon Group Relative Policy Optimization (GRPO) [46]. The "Region-Conditioned" aspect implies that $\pi_\theta$ is explicitly conditioned on the visual state, including dynamically incorporated regional evidence.

To estimate the advantage of each reasoning trajectory, we normalize its reward relative to the group as follow:

$$\hat{A}^i = \frac{r^i - \text{mean}(\{r^1, r^2, ..., r^M\})}{\text{std}(\{r^1, r^2, ..., r^M\})} \tag{1}$$

Here, $r^i$ is the total reward for the $i$-th trajectory in a group of $M$ trajectories, and $\hat{A}^i$ serves as a form of advantage function relative to the group performance.

A critical adaptation in R-GRPO concerns the computation of the policy gradient and the actions considered in the objective. In our interleaved image-text sequences, some tokens are generated by the model (textual reasoning, bbox commands), while others (the representations of cropped images) are injected by the environment. The policy gradient should only optimize the likelihood of actions generated by the model. Therefore, when calculating the gradient of $\log \pi_\theta(a_t|s_t)$, we apply a mask: the gradient is computed only for tokens $a_t$ that are text tokens or bounding box command tokens, masking out gradients for tokens corresponding to injected image regions. Conceptually, the sum of actions $\mathcal{A}_s$ in the loss primarily considers the probabilities of generating valid text tokens and bounding box commands, weighted by their advantage. The injected image tokens influence the state $s_{t+1}$ but are not actions $a_t$ for which we compute a policy gradient.

Following this, we optimize the policy model $\pi_\theta$ with the loss function defined as:

$$\mathcal{L}_{\text{GRPO}} = -\mathbb{E}_{Q \in D_S} \left[ \sum_{i=1}^{M} \frac{\pi_\theta(c^i|Q)}{\pi_\theta(c^i|Q)|_{\text{no grad}}} \hat{A}^i - \beta D_{KL}(\pi_\theta||\pi_{\text{ref}}) \right] \tag{2}$$

where $D_S$ is the dataset of question-state pairs, $Q$ represents a specific question and current visual state, $c^i$ is the sequence of generated tokens for the $i$-th trajectory given $Q$, and $\beta$ is a coefficient for the KL divergence term. The first term in the sum uses the normalized reward $\hat{A}^i$ to weight the likelihood of the generated sequence, encouraging sequences with higher relative rewards.

The KL divergence between the policy model and the reference model is estimated as in [46]:

$$D_{KL}(\pi_\theta||\pi_{\text{ref}}) = \frac{\pi_{\text{ref}}(c^i|Q)}{\pi_\theta(c^i|Q)} - \log \frac{\pi_{\text{ref}}(c^i|Q)}{\pi_\theta(c^i|Q)} - 1 \tag{3}$$

The total reward $r^i$ for a trajectory comprises four components that encourage desired VLM-R$^3$ behaviors:

- **Accuracy Reward ($r_{acc}$):** A terminal reward: 1 if the final answer is correct, 0 otherwise.

- **Format Reward ($r_{format}$):** A terminal reward: 1 if the output uses correct <answer> tags, 0 otherwise.

- **Region Validity Reward ($r_{region}$):** An intermediate reward of 0.5 for each valid, non-redundant bounding box generated (only when the answer is correct), capped at 0.5 per episode.

- **Reasoning Length Reward ($r_{length}$):** An intermediate reward of 0.001 per character for reasoning steps, capped at 0.25 per episode to prevent verbosity.

The total reward is computed as:

$$r_{overall} = r_{acc} + r_{format} + r_{length} + r_{region} \cdot \mathbb{I}(r_{acc} = 1.0) \tag{4}$$

By optimizing this objective, R-GRPO encourages the VLM to learn a policy that not only leads to correct final answers but also involves generating logical textual reasoning and strategically gathering the necessary visual evidence.

## 4 Experiments

### 4.1 Experiment Setting

We covers several public benchmarks. General visionlanguage understanding is measured on MME [14] and MMMU[66]; complex mathematical reasoning on MathVista [30] and MathVision [55]; scientific question answering on ScienceQA [31]; and document understanding on DocQA [33]. We also assess hallucination rates with HallucinationBench[15]. We evaluate our method against three categories of multimodal models. The first category consists of open-source baselines without explicit reasoning capability, including Qwen2.5-VL 7B [8] (also used as our primary baseline), InternVL2.5-8B [11], and LLaVA-Next 8B [22]. The second category comprises closed-source non-reasoning systems, represented by Gemini-2 Flash [1] and GPT-4o [36]. The third category contains models equipped with dedicated reasoning modules, namely LLaVA-CoT 11B [61] , Mulberry-Qwen2VL 7B [63], R1-onevision 7B [62]. To probe the upper bound of performance, we also compare our results with two larger closed-source models o1 [37]. Moreover, we also evaluate our model on four vision-centric benchmarks: MMVP [54], V* [60], HR-Bench [56], and MME-Realworld [68] to evaluate the model's capability in fine-grained visual understanding and spatial reasoning.

### 4.2 Dataset Details

Our supervised fine-tuning experiments used the VLIR dataset, which comprises 11,810 samples in total. As shown in figure 3, the distribution of crops per image exhibits considerable variation: 11,105 images contain a single crop, 607 images feature two crops, 68 images have three crops, 16 images include four crops, 8 images contain five crops, and 6 images have six or seven crops (3 each). These samples are drawn from five distinct source datasets: GQA (4,057 samples), TextVQA (3,267 samples), DocVQA (1,497 samples), InfographicsVQA (1,497 samples), and VSR (1,492 samples). We categorize the crops based on their relative size, defined as the ratio of the bounding box area to the total image area: "very small" (ratio < 0.05) accounts for 5,280 crops; "small" (0.05 $\leq$ ratio < 0.25) comprises 4,043 crops; "medium" (0.25 $\leq$ ratio < 0.5) includes 1,914 crops; and "large" (ratio $\geq$ 0.5) consists of 573 crops.

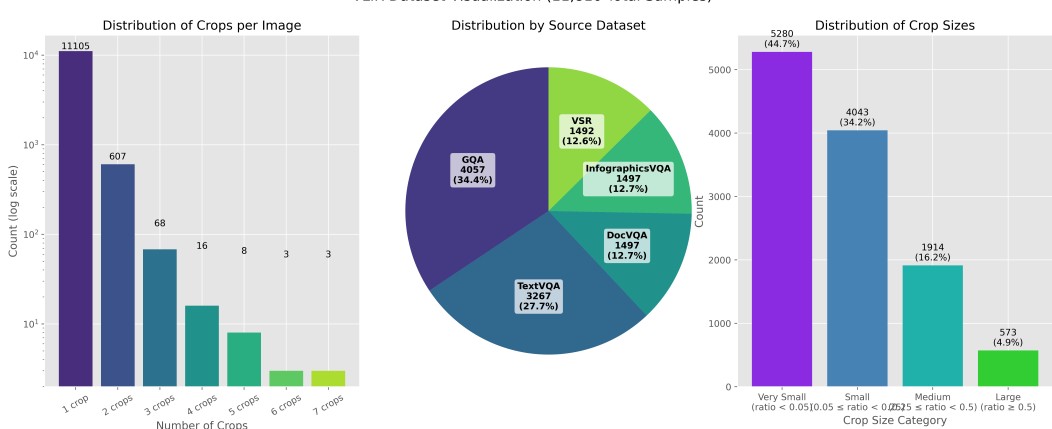

Figure 3: Distribution of the VLIR dataset: (a) number of crops per image, (b) samples across different source datasets, and (c) categorization of crops based on relative size.

Table 1: Performance comparison of various multimodal models across different benchmarks. Our model (in **bold**) is compared against non-reasoning models (both open-source and closed-source) and reasoning-based MLLMs. The highest values in each column are highlighted with green background.

| Model | Params | Benchmarks | | | | | | |
|---|---|---|---|---|---|---|---|---|
| | | MathVista | MathVision | MMMU | MME | ScienceQA | DocVQA | HallusionBench |
| *Closed-Source Non-Reasoning MLLMs* | | | | | | | | |
| Gemini-2 Flash | - | 73.1 | 41.3 | 71.7 | - | - | 92.1 | - |
| GPT-4o | - | 63.8 | 30.4 | 70.3 | 2328 | 66.2 | 91.1 | 56.2 |
| *Larger Closed-Source Models* | | | | | | | | |
| o1 | - | 71.8 | 63.2 | 77.6 | - | - | 81.6 | - |
| *Open-Source Non-Reasoning MLLMs* | | | | | | | | |
| InternVL2.5 | 8B | 58.3 | 17.1 | 51.8 | 2210 | - | - | - |
| LLaVA-Next | 8B | 37.5 | - | 41.7 | 1957 | 72.8 | - | - |
| *Open-Source Reasoning MLLMs* | | | | | | | | |
| LLaVA-CoT | 11B | 54.8 | - | - | - | - | - | 47.8 |
| R1-onevision | 7B | 64.1 | 29.9 | - | - | - | - | - |
| Vision-R1 | 7B | 73.5 | - | - | 2190 | - | - | 49.5 |
| Mulberry | 7B | 63.1 | - | 55.0 | - | - | - | 54.1 |
| Qwen2.5-VL | 7B | 68.2 | 25.1 | 58.6 | 2347 | 73.6 | 95.7 | 61.3 |
| **Ours** | 7B | **70.4** | **30.2** | **62.2** | **2432** | **87.9** | **96.8** | **62.0** |

## 4.3 Main Result

Our VLM-R$^3$ model, built upon the Qwen2.5-VL 7B architecture, consistently outperforms its base model across all benchmarks, with particularly significant gains in domains requiring precise visual reasoning and fine-grained understanding. Specifically, we observe a 2.2% improvement on Math-Vista (70.4% vs. 68.2%) and a remarkable 5.1% improvement on MathVision (30.2% vs. 25.1%), highlighting our method's effectiveness in mathematical reasoning tasks that demand careful attention to visual details. The substantial performance gain of 14.33% on ScienceQA (87.90% vs. 73.57%) further demonstrates VLM-R$^3$'s superior capability in scientific reasoning, where dynamic grounding of visual evidence is critical. When compared to other open-source reasoning-focused models like Vision-R1 and Mulberry, VLM-R$^3$ exhibits competitive performance on MathVista and surpasses Mulberry on HallusionBench (62.0% vs. 54.1%), indicating enhanced reliability in avoiding visual hallucinations. Our approach also narrows the gap with closed-source models like Gemini-2 Flash and o1, despite having significantly fewer parameters and being fully transparent in its architecture.

Table 2: Performance comparison and ablation study on vision-centric benchmarks. The highest values in each column are highlighted with green background.

| Model | MMVP | V* | HR-4k | HR-8k | MME-Real | Avg. |
|---|---|---|---|---|---|---|
| *Open-source General Models* | | | | | | |
| LLaVA-OneVision-72B | – | 73.8 | 66.3 | 60.9 | 48.7 | – |
| InternVL3-8B | – | 72.3 | 70.8 | 62.0 | 47.9 | – |
| InternVL3-38B | – | 77.5 | 76.3 | 67.0 | 51.0 | – |
| Qwen2.5VL-7B (Base) | 66.7 | 74.3 | 69.8 | 64.6 | 42.3 | 63.5 |
| Base + Vanilla Text RL | 72.0 | 78.5 | 72.9 | 64.1 | 46.2 | 66.7 |
| Base + Text-only Bbox | 72.4 | 79.5 | 73.2 | 65.2 | 47.2 | 67.5 |
| Base + Re-insert Full Image | 73.9 | 77.8 | 72.5 | 64.8 | 45.8 | 67.0 |
| **Ours** | **75.0** | **83.8** | **73.4** | **66.8** | **51.6** | **70.1** |
| $\Delta$ vs. Vanilla Text RL | +3.0 | +5.3 | +0.5 | +2.7 | +5.4 | +3.4 |
| $\Delta$ vs. Text-only Bbox | +2.6 | +4.3 | +0.2 | +1.6 | +4.4 | +2.6 |
| $\Delta$ vs. Re-insert Full Image | +1.1 | +6.0 | +0.9 | +2.0 | +5.8 | +3.1 |

Table 3: Ablation study on MathVista, MMMU, ScienceQA, and DocVQA benchmarks. We evaluate the contribution of each key component: Interleaved Chain-of-Thought, VLIR fine-tuning, and R-GRPO.

| Model Variant | MathVista | MMMU | ScienceQA | DocVQA | Avg. |
|---|---|---|---|---|---|
| Base Model (Qwen2.5-VL) | 68.2 | 58.6 | 73.6 | 95.7 | 74.0 |
| w/o Interleaved Chain-of-Thought | 67.1 ($\downarrow$3.3) | 59.4 ($\downarrow$2.8) | 75.4 ($\downarrow$12.5) | 95.9 ($\downarrow$0.9) | 74.4 ($\downarrow$4.9) |
| w/o VLIR Fine-tuning | 65.8 ($\downarrow$4.6) | 57.0 ($\downarrow$5.2) | 72.2 ($\downarrow$15.7) | 93.3 ($\downarrow$3.5) | 72.1 ($\downarrow$7.2) |
| w/o R-GRPO | 69.7 ($\downarrow$0.7) | 60.8 ($\downarrow$1.4) | 84.6 ($\downarrow$3.3) | 96.1 ($\downarrow$0.7) | 77.8 ($\downarrow$1.5) |
| Full VLM-R$^3$ (Ours) | **70.4** | **62.2** | **87.9** | **96.8** | **79.3** |

Table 2 presents the performance of our VLM-R$^3$ model on vision-centric benchmarks, including MMVP [54], V* [60], HRBench [56], and MME-Realworld [68]. Our model shows superior capability in handling complex visual reasoning tasks. Notably, VLM-R$^3$ attains an average score of 70.1%, significantly outperforming the base Qwen2.5-VL model (63.5%) and larger open-source models like LLaVA-OneVision-72B [20] and InternVL3-38B [71].

## 4.4 Ablation Study

To assess the contribution of each component in our VLM-R$^3$ framework, we conduct comprehensive ablation experiments. Table 2 and Table 3 summarize our findings.

### 4.4.1 Effectiveness of Interleaved Chain-of-Thought

To isolate the impact of our interleaved reasoning approach, we conduct an experiment where we maintain the region localization capabilities (bounding boxes) but remove the associated region images from the reasoning chain. This variant relies solely on textual descriptions of identified regions without visually grounding each reasoning step. As shown in Table 3, removing the interleaved visual evidence leads to a consistent performance drop across all benchmarks, with particularly notable decreases on ScienceQA (-12.5%) and MMMU (-2.8%). This degradation is most pronounced in tasks requiring fine-grained visual understanding, such as scientific diagrams in ScienceQA, where purely textual descriptions of regions fail to capture crucial visual patterns and spatial relationships.

In Table 2, we further compare this variant against other ablations on vision-centric benchmarks. The results indicate that the interleaved reasoning chain achieves an average improvement of 3.4% over a vanilla text-based reinforcement learning approach. Furthermore, it outperforms variants that rely solely on text-based bounding box descriptions (+2.6%) or re-inserting the full image at each reasoning step (+3.1%). These findings underscore the critical role of dynamically incorporating visual regions into the reasoning process for effective multimodal understanding.

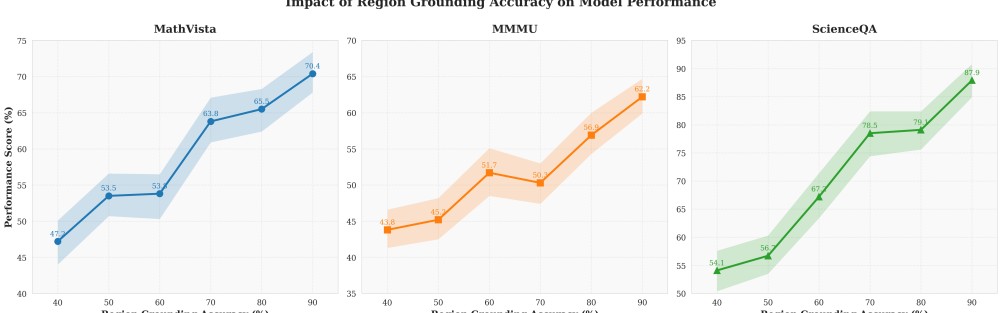

Figure 4: Impact of region grounding accuracy on model performance across three benchmarks. Each subplot shows the performance trajectory from 40% to 90% grounding accuracy with confidence intervals (shaded regions).

### 4.4.2 Effectiveness of Finetuning on VLIR

Our approach leverages the VLIR corpus to bootstrap the model's ability to identify informative regions and incorporate them into coherent reasoning chains. To evaluate the specific contribution of VLIR fine-tuning, we experiment with a variant that skips this initialization phase and proceeds directly to R-GRPO training. The results in Table 3 demonstrate that omitting VLIR fine-tuning leads to performance degradation across all benchmarks, with particularly significant decreases observed in ScienceQA (-15.7%) and MMMU (-5.2%). More critically, we observed that ablating VLIR fine-tuning impairs the model's instruction-following capabilities, leading to substantial deficiencies such as failures to adhere to required formatting conventions for bounding box specifications. This accounts for the substantial performance deterioration observed in our experimental results.

### 4.4.3 Effectiveness of R-GRPO

To assess the impact of our Region-Conditioned Reinforcement Policy Optimization (R-GRPO), we evaluate a variant that relies solely on supervised fine-tuning using the VLIR corpus without the subsequent reinforcement learning stage. This allows us to isolate the specific benefits of our reinforcement learning approach over purely supervised learning. The experimental results show that removing R-GRPO reduces performance in all benchmarks, with the highest decreases observed in ScienceQA (-3.28%) and MathVista (-0.7%). This suggests that while VLIR fine-tuning provides a strong foundation, the reinforcement learning stage is essential for optimizing the model's region selection and reasoning policies beyond what can be achieved through imitation learning alone.

### 4.5 Discussion

### 4.5.1 Impact of Region Grounding Accuracy on the Reasoning Chain

The quality of region grounding, represented by the accuracy of bounding boxes (bbox), plays a critical role in multimodal reasoning capabilities. Our analysis investigates how varying levels of grounding accuracy impact the performance of the VLM-R$^3$ model across multiple benchmarks. We systematically evaluated model performance by controlling grounding accuracy from 40% to 90% and measuring outcomes on three key benchmarks: ScienceQA, MathVista, and MMMU. Grounding accuracy was manipulated by randomly replacing or perturbing a controlled percentage of bounding boxes in the input. As shown in Figure 4, there is a clear positive correlation between region grounding accuracy and model performance across all three benchmarks. ScienceQA demonstrates the most substantial improvement, with performance increasing from 54.1% at 40% grounding accuracy to 87.9% at 90% grounding accuracy. MathVista shows a similar upward trend, rising from 47.9% to 70.4%, while MMMU exhibits more modest but consistent gains from 43.8% to 62.2%. These results underscore the fundamental importance of precise region grounding for effective multimodal reasoning, with higher-level reasoning tasks showing greater sensitivity to grounding quality.

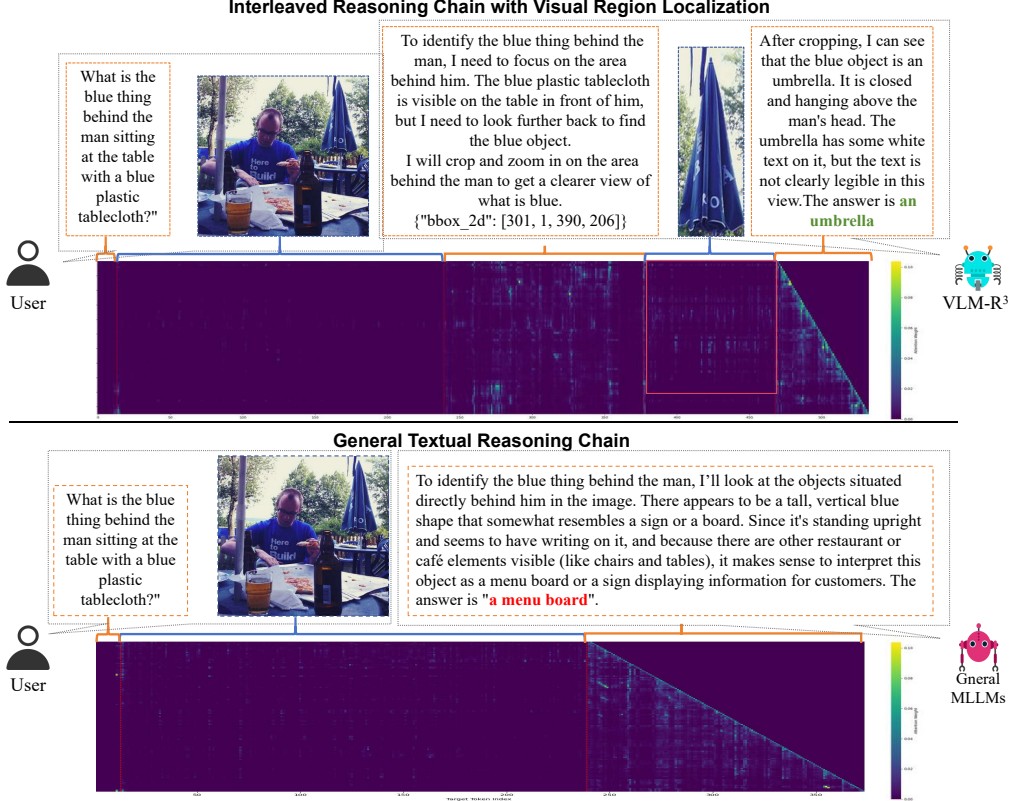

Figure 5: Comparison of attention distribution patterns between the interleaved reasoning chain with visual region localization (top) and general textual reasoning chain (bottom).

### 4.5.2 Why is the Interleaved Reasoning Chain with Visual Region Localization Effective?

To understand the efficacy of our VLM-R[3] approach, we conducted a comparative analysis between the interleaved reasoning chain with visual region localization and traditional textual reasoning chains. Figure 5 visualizes the attention distribution patterns for both approaches when answering the same visual query. Our analysis reveals a critical insight: in traditional approaches where the image is positioned at the beginning of the sequence, attention to visual information diminishes significantly as the reasoning chain progresses. As shown in the lower portion of Figure 4, general MLMs tend to make incorrect inferences (identifying a "menu board" instead of an umbrella) as they lose visual context during extended reasoning. In contrast, VLM-R[3] maintains persistent visual attention throughout the reasoning process by dynamically localizing and incorporating relevant visual regions. The attention heatmap demonstrates that tokens generated later in the reasoning process maintain strong attention connections to the cropped visual regions. This region-specific attention enables the model to correctly identify the blue object as an umbrella by explicitly focusing on the area behind the person, cropping it for detailed examination, and making accurate observations about its features.

## 5 Conclusion

This paper introduced VLM-R[3], a novel framework enabling MLLMs to perform dynamic visual reasoning through region recognition, reasoning, and refinement. By integrating our custom VLIR dataset and Region-Conditioned Reinforcement Policy Optimization (R-GRPO), we demonstrated that interleaved visual-textual chains-of-thought significantly outperform traditional approaches. VLM-R[3] achieves state-of-the-art results across multiple benchmarks, particularly excelling in tasks requiring fine-grained spatial reasoning and visual evidence integration. Our work opens promising directions for developing more sophisticated visually-grounded reasoning systems that can adaptively focus on relevant regions during multi-step inference processes.

# 6 Acknowledgement

This work is supported by the National Natural Science Foundation of China (NO.623B2007), Chinese Institute of Electronics (CIE) - Tencent Doctoral Research Incentive Program (Hunyuan Large Language Model Special Track) (NO.841010445) and CCF-Zhipu Large Model Innovation Fund (NO.CCF-Zhipu202415).

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

# A  Experiment Settings

## A.1  Pipline Settings

### A.1.1  Model Hyperparameter Settings

Our base model is Qwen2.5VL-7B[8], which supports dynamic resolution for input images. In all experiments, we constrained the pixel dimensions of each image to a minimum of 3136 pixels and a maximum of 1605632 pixels. Because the value of the bounding box is related to the number of pixels in the input image, the setting of the range of pixels needs to be unified.

### A.1.2  Zoom Scaling Rule

In our pipeline, when a region is selected for closer inspection (e.g., via a "Crop" operation), a zoom operation is applied. The scaling factor for this zoom, denoted as scale, is determined dynamically based on the relative area of the selected bounding box ($A_{\text{bbox}}$) compared to the area of the original image ($A_{\text{orig}}$). Let $r = \frac{A_{\text{bbox}}}{A_{\text{orig}}}$ be this area ratio. The scale is calculated using the following piecewise function:

$$\text{scale} = \begin{cases} 2.0, & \text{if } r < 0.125 \\ 1.0, & \text{if } r \geq 0.5 \\ 2.0 - \dfrac{r - 0.125}{0.375}, & \text{otherwise} \end{cases} \tag{5}$$

This rule implies that smaller selected regions (smaller $r$) are scaled up more significantly (up to a factor of 2.0), while larger regions (larger $r$) are scaled up less, or not at all if they already occupy a substantial portion of the original image. The intermediate case provides a linear interpolation of the scaling factor.

## A.2  Training Setting for Supervised Fine-tuning Stage

In the supervised fine-tuning stage, we used the complete VLIR dataset. Our experiments were conducted on 4 NVIDIA A100 GPUs, each equipped with 80GB of memory, leveraging DeepSpeed[42] for efficient training. We used a batch size of 2 with a gradient accumulation of 8, a learning rate of $2 \times 10^{-7}$, and trained for 3 epochs. During this phase, the vision encoder and MLP projector were frozen, and only the Large Language Model (LLM) component was trained.

## A.3  Training Setting for R-GRPO Stage

For the R-GRPO stage, we sampled approximately 5,000 data points from TextVQA [48], GQA [17], VSR [26], DocVQA [33] and M$^3$CoT [9] datasets. Regarding the hyperparameters for the GRPO formulation(2), we set $M = 5$. Following the experience of related studies, we set $\beta = 0.0$, i.e., we eliminate the KL divergence constraint.

Our experiments for R-GRPO were performed on 6 NVIDIA A100 GPUs, each with 80GB of memory, also utilizing DeepSpeed[42]. The batch size per device was set to 1, with a gradient accumulation of 16. The learning rate was $1 \times 10^{-6}$, and training continued for 300 steps. We employ a rule-based reinforcement learning approach, where the correctness of the final answer was judged using an exact match criterion. Similar to the supervised fine-tuning stage, the vision encoder and MLP projector were frozen, and only the LLM component was trained.

# B  Prompt Templates for VLIR Dataset Construction and Filtering

## B.1  Data Construction Prompts

Given an {image, question, answer} triplet, the following prompt was used to construct the interleaved visual-linguistic chain of thought:

```
You are performing "Multimodal Interleaved Reasoning". During the thinking
```

```
process, you need to keep an eye on the visual cues in the original image,
find regions of the image that help answer the question, and use the "Crop"
tool to crop and zoom in for detailed analysis.
When using the tool, you must output a JSON object in the following format:
{"bbox_2d": [x1, y1, x2, y2]}
Ensure that you "Crop" at least once.
Continue thinking after each operation until you reach the final answer.
Output the thinking process within a pair of <think> </think> tags and then
output the final answer within a pair of <answer> </answer> tags.
{question}
```

Listing 1: Prompt for dataset construction.

Given an {image, question, answer, bounding box annotation} quadruplet, the following prompt was used:

```
I will now provide you with an image, a question, and a "Crop" operation
string. Your task is to write the reasoning process used to answer the
question as instructed. During the reasoning process, the respondent
utilizes a "Crop" operation to assist with reasoning. The format of
the operation is as follows:
{"bbox_2d": [x1, y1, x2, y2]}
This bounding box indicates the key region that needs to be focused
on to correctly answer the question.
You must think step by step from the perspective of the respondent,
using the "Crop" operation at appropriate moments in your reasoning
process to eventually reach the correct answer. Important notes:
1. You must not modify the content or format of the "Crop" operation
in any way.
2. In a real setting, the respondent only has access to the image and
the question. This bounding box indicates the area where the correct
answer information is located. In this task, they are provided to ensure
the correctness of your reasoning process. When writing the reasoning,
pretend you are the respondent who independently identifies when to use
the "Crop" operation and how to reach the answer step by step.
3. Make sure the reasoning is fluent, logical, and concise.
4. Format of the reasoning process: <think>...</think><answer>...</answer>

Here is an example:
Question: Are there any black numbers or letters?
"Crop" operation: {"bbox_2d": [247, 384, 307, 444]}
Reasoning: <think>
Step 1: To determine if there are black numbers or letters, I need to
focus on the text visible in the image. The dog is wearing a heart-shaped
tag that has some text on it. I will crop and zoom in on the tag for a
closer look at the text details. {"bbox_2d": [247, 384, 307, 444]}
Step 2: After cropping, I can see that the letters "G PLUS" are in red,
and the numbers "6 223 13" are also in red. There are no black numbers
or letters on the tag. Review the rest of the image, there are no black
numbers or letters either.</think>
<answer>no</answer>

Question:{question}
"Crop" operation:{crop}
Now Output the reasoning process:
```

Listing 2: Prompt for dataset construction.

## B.2 Data Filtering Prompts

The prompt for assessing the recognizability of the cropped images is as follows:

```
You need to determine whether the content in a picture is a complete and
semantically meaningful visual unit. Please look carefully at this cropped
image and determine whether it contains a recognizable object, block of text,
or specific part of a diagram. If it is recognizable, answer 'yes'; if not,
answer 'no'.
Now output 'yes' or 'no' directly.
```
Listing 3: Prompt for assessing cropped image recognizability.

The prompt for assessing the quality of the reasoning process is as follows:

```
You need to make an in-depth assessment of this reasoning process. First,
determine whether its logic is rigorous and whether each step of reasoning leads
naturally and smoothly to the next; second, check whether the reasoning process
progresses gradually towards arriving at the final answer; and lastly, check
whether there is any false information or repetitive redundancy in the text
that is not relevant to the reasoning. If this textual reasoning meets the
requirements in terms of logic, advancement and content streamlining, output
'yes'; whenever one of these is not met, output 'no'.
{question}
{ground-truth answer}
{reasoning process}
Now output 'yes' or 'no' directly.
```
Listing 4: Prompt for assessing reasoning process quality.

## C    More Demonstrations

This section provides qualitative examples illustrating the capabilities of VLM-R[3].

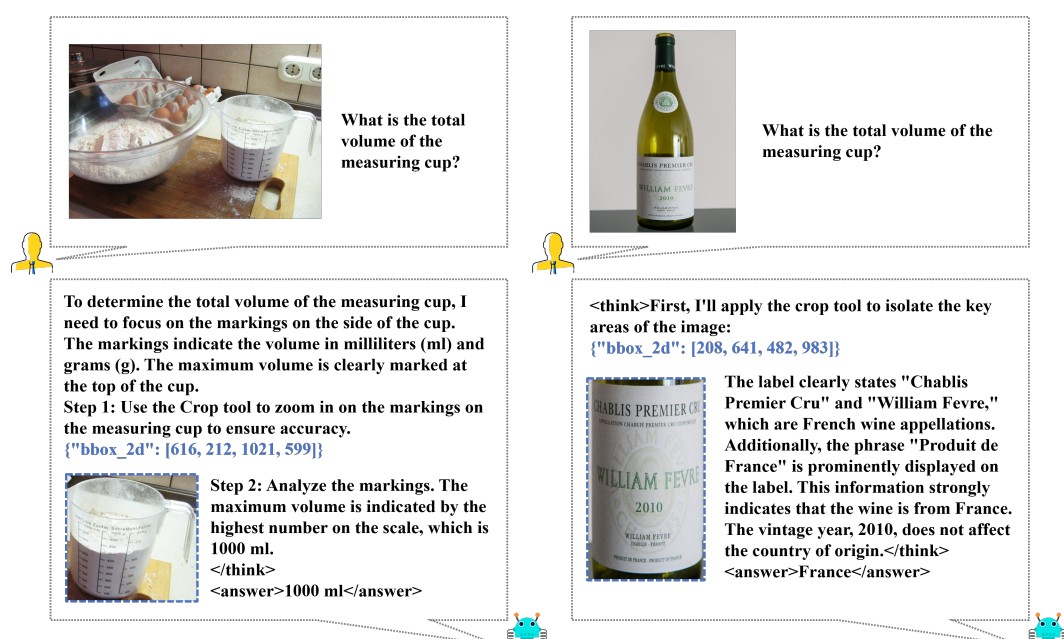

Figure 6: This figure presents how VLM-R[3] captures details in a natural image and reasons about them.

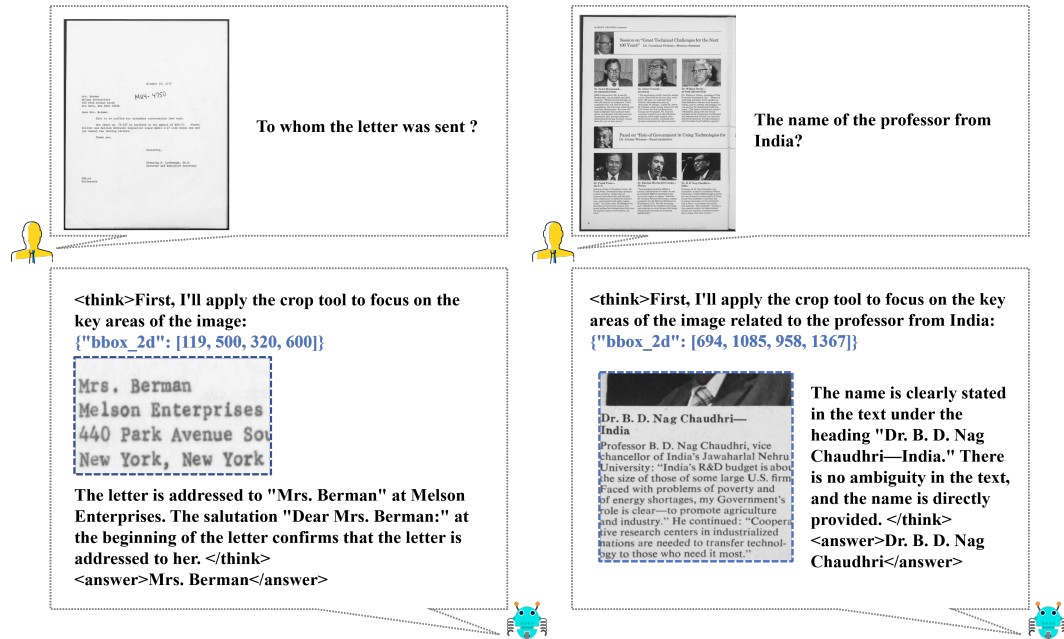

Figure 7: This figure presents how VLM-R$^3$ gets the text details from the document image.

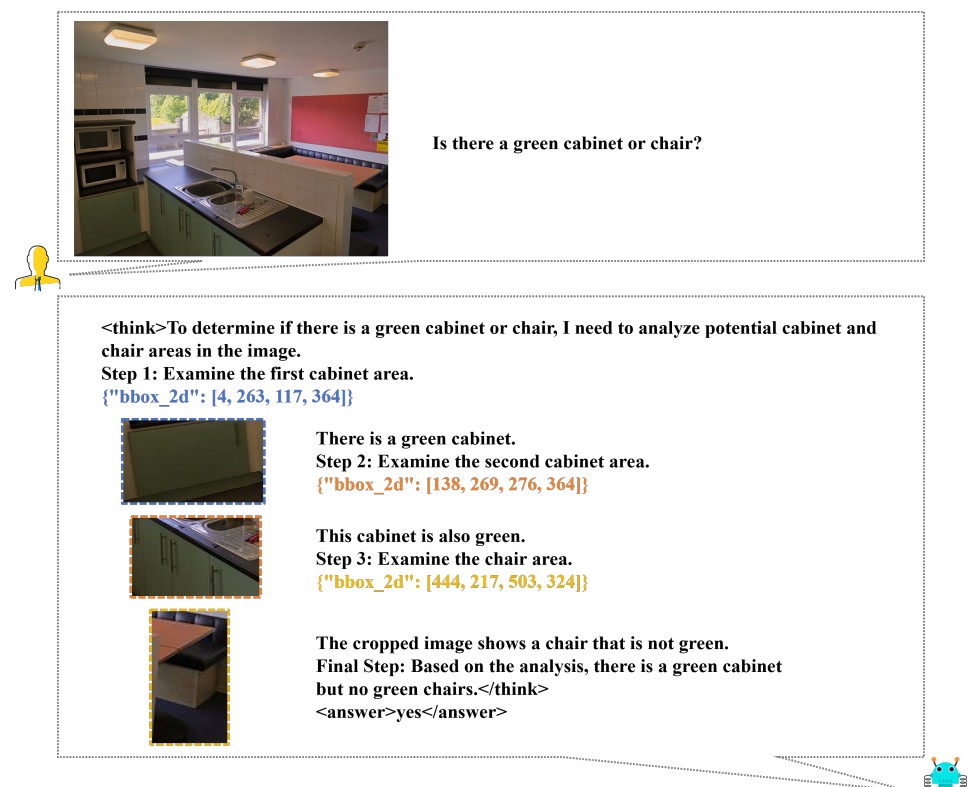

Figure 8: This figure presents how VLM-R$^3$ iteratively tracks multiple visual cues in an image.

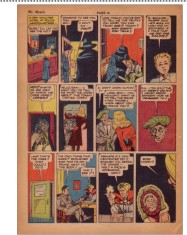
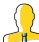

**What old comic book is this?**

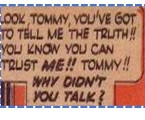
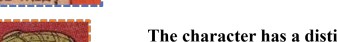
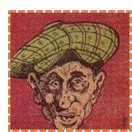

<think>To determine the old comic book, I need to identify key elements such as the title, characters, and style.
Step 1. Zoom in on the title text for better clarity.
{"bbox_2d": [377, 51, 553, 126]}

The cropped image shows part of the dialogue but not the title. Let's try another approach.
Step2: Zoom in on the main character for better analysis.
{"bbox_2d": [559, 489, 688, 617]}

The character has a distinctive appearance with a hat and a unique face. This style is reminiscent of older comics.
Based on the title "Mr. Mystic," the page number, and the art style, this comic appears to be an old issue of "Mr. Mystic."</think>
<answer>This is an old comic book titled Mr. Mystic.</answer>

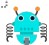

Figure 9: This figure presents how VLM-R$^3$ performs complex interleaved text-image CoT reasoning.

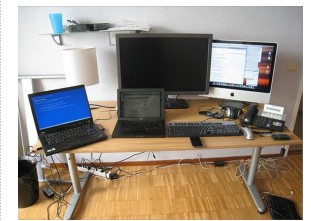
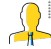

**What can be inferred from the objects on the desk? Choose from the following options:**
**A. The computer owner is a game player**
**B. The computer owner is an editor**
**C. The computer owner has a diverse repertoire of computer knowledge**
**D. The computer owner is forgetful**

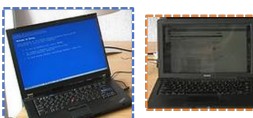
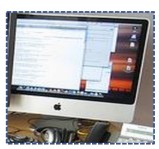

<think>To determine the correct inference about the computer owner, I need to analyze the objects on the desk. The presence of multiple monitors, laptops, and various peripherals suggests a setup that could be used for different types of work or activities. Let's focus on the key areas of the image to gather more specific details.I crop and zoom in on the laptops and monitors to identify any specific software or applications that might indicate the owner's profession or activities.
{"bbox_2d": [17, 131, 156, 255]}  {"bbox_2d": [168, 139, 262, 222]}

The cropped images show two laptops. The first laptop has a blue screen with text, which could indicate a boot-up screen or an error message. The second laptop has a blank screen, which could mean it is not in use or is in a sleep mode. Now, let's look at the monitors. {"bbox_2d": [336, 57, 462, 186]}

The first monitor is turned off, and the second monitor is displaying a desktop with various open applications, which could indicate that the owner is engaged in multitasking or working on multiple projects simultaneously.Based on the analysis, the owner seems to have a diverse setup that could be used for various tasks, such as programming, editing, or general computing. However, there is no specific evidence to suggest that the owner is a game player, an editor, or forgetful. The setup is more indicative of someone who has a broad range of computer knowledge and is likely to be engaged in multiple activities.</think>
<answer>C</answer>

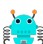

Figure 10: This figure presents how VLM-R$^3$ performs complex interleaved text-image CoT reasoning.

