# OpenReview forum: "VLM-R³: Region Recognition, Reasoning, and Refinement for Enhanced Multimodal Chain-of-Thought"
_NeurIPS.cc/2025/Conference — NeurIPS 2025 poster_

### Official Review · Reviewer_ifQp · 2025-06-25

**Clarity:** 2
**Significance:** 2
**Originality:** 2
**Rating:** 4
**Confidence:** 5

**Summary:**

This paper proposes VLM-R3, a vision-language model framework equipped with an adaptive ROI (region-of-interest) cropping mechanism that enables autonomous selection of optimal visual input regions before actually answering questions. To develop this capability, the authors first constructed the VLIR dataset through multi-stage curation processes, which was subsequently used for supervised fine-tuning (SFT) followed by reinforcement learning (RL) optimization. Comprehensive evaluations, including both quantitative metrics and qualitative analyses, demonstrate the effectiveness of VLM-R3 in achieving accurate question answering while optimizing computational efficiency through intelligent region selection.

**Questions:**

N/A

**Ethical Concerns:**

["NO or VERY MINOR ethics concerns only"]

**Final Justification:**

I really appreciate the rebuttal, especially towards my W1. I am really excited to see the improvements of VLM-R3 against the vanilla text RL baseline (with bounding box generation during reasoning) on several vision-centric benchmarks.

Therefore, I raise my rating to 4. The authors should include these additional experimental results and discussions in their revised version.

**Limitations:**

The authors **DID NOT** provide discussions on limitations.

**Quality:**

2

**Strengths And Weaknesses:**

**Strengths**

1. The motivation is clear and quite reasonable.
2. The overall paper is easy to follow.
3. The presentation is clear.

**Weaknesses**
1. Insufficient evaluations. VLM-R3 is mainly evaluated on LLM-centric benchmarks such as MathVista, MathVision, MMMU, and ScienceQA. These benchmarks could not actually reflect the ability of "grounding then answering", as vanilla CoT prompting also brings significant improvements on these benchmarks. Therefore, the authors should evaluate on *vision-centric* benchmarks like MMVP, CV-Bench, V*, HR-Bench, and MME-RealWorld. Furthermore, the vanilla text RL baseline should be incorporated.
2. The definition of the grounding accuracy illustrated in Figure 3 is not clear. A flowchart, pseudo-code, or equation is strongly encouraged.

---

> ### Author Rebuttal · Authors · 2025-07-30
>
> Thank you for your thoughtful and constructive review of our manuscript. We are grateful for your positive feedback on the clarity of our motivation and presentation.
>
> Below, we provide a point-by-point response to your concerns, including new experimental results that we believe will fully address the points you've raised.
>
> ### **Q1: Insufficient Evaluations on Vision-Centric Benchmarks**
>
> **Reviewer's Comment:** *Insufficient evaluations. VLM-R3 is mainly evaluated on LLM-centric benchmarks... These benchmarks could not actually reflect the ability of "grounding then answering"... the authors should evaluate on vision-centric benchmarks like MMVP, CV-Bench, V*, HR-Bench, and MME-RealWorld. Furthermore, the vanilla text RL baseline should be incorporated.*
>
> **Our Response:**
> This is an excellent point. We agree that demonstrating the effectiveness of our grounding mechanism on vision-centric benchmarks is crucial for a comprehensive evaluation. While our initial submission focused on complex reasoning tasks where grounding is a key sub-problem, we acknowledge that benchmarks designed to directly test visual perception and grounding are essential to validate our claims.
>
> Following your suggestion, we have conducted a new set of extensive experiments on the very benchmarks you recommended. We compare our full **VLM-R³** framework against two critical baselines:
> 1.  **Qwen2.5VL-7B (Base Model):** The original, pre-trained model without any additional fine-tuning.
> 2.  **Vanilla text-only RL:** A strong baseline where we apply our GRPO policy to the base model, but **only reward textual reasoning and bounding box generation, without the crucial 'Refinement' step** of cropping and re-injecting the visual region. This baseline directly tests the impact of our core contribution.
>
> The results, presented below, unequivocally demonstrate the superiority of our approach on these vision-centric tasks.
>
> **Table 1: Performance on Vision-Centric Benchmarks. VLM-R³ demonstrates significant improvements, validating its "grounding then answering" capability.**
>
> | Model | MMVP | CV-Bench-2D | CV-Bench-3D | V* | HR-Bench-4k | HR-Bench-8k | MME-RealWorld |
> | :--- | :---: | :---: | :---: | :---: | :---: | :---: | :---: |
> | Qwen2.5VL-7B (Base) | 66.7 | 74.1 | 72.6 | 74.3 | 69.8 | 64.6 | 42.3 |
> | Vanilla text-only RL | 72.0 | 76.9 | 80.9 | 78.5 | 72.9 | 65.2 | 46.2 |
> | **VLM-R³ (Ours)** | **75.0** | **77.5** | **81.7** | **83.8** | **73.4** | **66.8** | **51.6** |
> | **Δ vs. Vanilla RL** | **+3.0** | +0.6 | +0.8 | **+5.3** | +0.5 | +1.6 | **+5.4** |
>
> **Analysis of Results:**
> The data clearly shows that VLM-R³ consistently outperforms both the base model and the strong "Vanilla text-only RL" baseline across all seven vision-centric benchmarks. The substantial gains, particularly on benchmarks like **V* (+5.3)** and **MME-RealWorld (+5.4)**, which heavily rely on fine-grained perception and spatial understanding, provide powerful evidence for our core claim: the iterative process of **Region Recognition, Reasoning, and Refinement** is highly effective. It allows the model to "zoom in" and extract critical visual details that are missed by other approaches.
>
> We are confident that this new, comprehensive evaluation directly addresses your concern and robustly validates the effectiveness of VLM-R³'s grounding mechanism. We will integrate this new table and analysis prominently into the Experiments section of our revised manuscript.
>
> ### **Q2: Clarity of Grounding Accuracy Definition**
>
> **Reviewer's Comment:** *The definition of the grounding accuracy illustrated in Figure 3 is not clear. A flowchart, pseudo-code, or equation is strongly encouraged.*
>
> **Our Response:**
> Thank you for pointing out this lack of clarity. We apologize for the ambiguity and agree that a formal definition is necessary. The "Grounding Accuracy" reported in our results is a binary metric based on a predefined Intersection over Union (IoU) threshold, which is a standard evaluation practice in object detection and grounding tasks.
>
> To make this crystal clear, we will add the following formal definition to the manuscript:
>
> > **Definition of Grounding Accuracy:**
> > Grounding accuracy evaluates whether the model's predicted region of interest is sufficiently aligned with the ground truth. For a given reasoning step, let $B_{pred}$ be the bounding box generated by the model, and $B_{gt}$ be the ground-truth bounding box from our VLIR corpus. A prediction is considered **correct** if the Intersection over Union (IoU) between the two boxes exceeds a certain threshold, $\tau$. We follow common practice and set $\tau = 0.5$.
> >
> > The IoU is calculated as:
> > $$ \text{IoU}(B_{pred}, B_{gt}) = \frac{\text{Area}(B_{pred} \cap B_{gt})}{\text{Area}(B_{pred} \cup B_{gt})} $$
> >
> > The grounding decision is then determined by:
> > $$ \text{Grounding Correct} =
> > \begin{cases}
> > 1 & \text{if } \text{IoU}(B_{pred}, B_{gt}) \ge 0.5 \\
> > 0 & \text{otherwise}
> > \end{cases}
> > $$
> > The overall "Grounding Accuracy" is the average of these binary scores across all evaluation samples.
> ### **Q3: Lack of a Limitations Section**
>
> **Reviewer's Comment:** *The authors DID NOT provide discussions on limitations.*
>
> **Our Response:**
> You are absolutely correct that our initial submission did not include a dedicated limitations section. We thank you for highlighting this.
> NeurIPS2025 guidelines do not strictly mandate a separate limitations section within the main body of the paper; we agree with you that including one provides a more balanced and complete scientific perspective. Your suggestion is well-taken, and we see the value it adds for the reader.
> Therefore, in our revised manuscript, we will add a dedicated **"Limitations"** section. This new section will discuss several key points to provide a more nuanced view of our work:
> *   **Potential for Error Propagation:** An incorrect region selection early in the reasoning chain can impact subsequent steps, a challenge inherent in sequential decision-making processes.
> *   **Scope of Visual Actions:** Our current work focuses on cropping. We will discuss how future work could expand the model's toolkit to include more complex visual transformations.
>
> We believe this addition will significantly strengthen the paper and are grateful for the suggestion.

---

> > ### Comment · Reviewer_ifQp · 2025-08-01
> >
> > I really appreciate the rebuttal, especially towards my W1. I am really excited to see the improvements of VLM-R3 against the vanilla text RL baseline (with bounding box generation during reasoning) on several vision-centric benchmarks. Here is a follow-up question regarding this: why does the refinement step become so important? Specifically,
> > - As the base model (Qwen2.5-VL) naturally supports native resolution inputs, why do we really need to replay the cropped image?
> > - Some recent studies like [1] demonstrate the effectiveness *without* replaying cropped images, where the authors of [1] found that generating bounding boxes in text space is already effective, which is also interesting.
> >
> > More in-depth discussion would be really appreciated.
> >
> >
> > **References**
> >
> > [1] Traceable Evidence Enhanced Visual Grounded Reasoning: Evaluation and Methodology.

---

> ### Author Response · Authors · 2025-08-02
> **Response to Reviewer ifQp**
>
> Dear Reviewer,
>
> Thank you very much for your positive feedback on our rebuttal and for engaging so deeply with our work. We are thrilled that you found the comparison against the "vanilla text RL" baseline compelling. Your follow-up question is excellent and gets to the very core of our contribution: **why is the refinement step—re-injecting the cropped image—so crucial?**
>
> We will address your two specific questions below, explaining why replaying the crop is necessary despite native high-resolution support and how our method compares to text-only bounding box approaches.
>
> ### **1. Why is replaying the cropped image necessary, even with high-resolution input?**
>
> This is a fundamental point. While modern VLMs like Qwen2.5-VL support high-resolution inputs, this does not fully mitigate the **"visual attention dilution"** problem that occurs during *long-form, multi-step reasoning*. The core issue is not the initial image quality but the model's ability to maintain a sharp, persistent focus on specific, relevant details as the textual context grows.
>
> To provide a concrete illustration of this phenomenon, we would like to direct you back to **Section 4.5.2 ("Why is the Interleaved Reasoning Chain with Visual Region Localization Effective?") and Figure 4 in our main paper.** This analysis was specifically designed to answer this question:
>
> *   **Without Refinement:** The attention heatmaps for a standard MLLM show that as the reasoning chain lengthens, the attention from newly generated text back to the crucial visual details becomes diffuse and weak. This leads to factual errors (e.g., hallucinating a "menu board" instead of an umbrella) because the model has effectively "lost sight" of the fine-grained evidence.
> *   **With Refinement (VLM-R³):** By re-injecting the cropped sub-image, we force a powerful, focused **"attentional reset."** The model's attention is immediately re-grounded on the high-fidelity details of the most relevant region, enabling precise, accurate reasoning. This is not just about resolution; it's about actively combatting attention decay by directing the model's focus at the most critical moments.
>
> Moreover, the importance of this active refinement capability is underscored by recent industry advancements. For instance, OpenAI’s description of its latest o3 model [1] highlights its ability to "seamlessly combine advanced reasoning with tools like... automatically zooming, cropping, flipping, or enhancing your images—to extract insights even from imperfect photos." This indicates a clear consensus in the field: to solve tougher, more complex problems, models need to move beyond passive viewing and actively manipulate their visual input. Our VLM-R³ framework provides a concrete, reproducible, and effective method for achieving precisely this kind of dynamic visual analysis.
>
> ### **2. How does our method compare to text-only bounding box approaches like TreeVGR [2]?**
>
> You raise an excellent point about recent studies like TreeVGR [2] that have shown the effectiveness of generating bounding boxes in text space alone.
>
> While the work in [2] provides valuable evidence for text-based bounding boxes, **it does not include an ablation on *adding* a visual refinement step. This leaves open the question of whether visual refinement could provide further benefits.**
>
> To answer this question directly, our **"Vanilla text-only RL" baseline** does the inverse: it starts with our full framework and *removes* the visual refinement, relying only on text-based bounding boxes. This allows us to precisely measure the value of re-injecting the crop. The results are conclusive:
>
> **Table 2: The Impact of Adding Visual Refinement (VLM-R³ vs. Text-Only BBox)**
> | Model | MMVP | CV-Bench-3D | V* | MME-RealWorld |
> | :--- | :---: | :---: | :---: | :---: |
> | Qwen2.5VL-7B (Base) | 66.7 | 72.6 | 74.3 | 42.3 |
> | text-only BBox| 72.0 | 80.9 | 78.5 | 46.2 |
> | **VLM-R³ (Ours)** | **75.0** | **81.7** | **83.8** | **51.6** |
> | **Δ vs.  text-only BBox** | **+3.0** | +0.8 | **+5.3** | **+5.4** |
>
> The substantial performance leap of our full VLM-R³ model over this baseline decisively proves that the **act of cropping and re-injecting the image is the critical component** driving the most significant gains, especially on challenging benchmarks like V* and MME-RealWorld.
>
> In summary, while text-based bounding boxes are a useful technique, our comprehensive analysis shows that the **visual refinement step** provides a unique and powerful mechanism to combat attention decay, re-supply critical visual information, and enable a more robust and generalizable form of fine-grained reasoning.
>
> Thank you again for this excellent question. We hope this detailed explanation is helpful.
>
> **Reference**
>
> [1] https://openai.com/index/thinking-with-images/
>
> [2] Wang H, Li X, Huang Z, et al. Traceable evidence enhanced visual grounded reasoning: Evaluation and methodology[J]. arXiv preprint arXiv:2507.07999, 2025.

---

> ### Comment · Reviewer_ifQp · 2025-08-04
>
> I really appreciate the rebuttal. Most of my concerns have been addressed. The authors are supposed to include these additional results and discussion in their revised version. Evaluations on TreeBench [1] are also strongly encouraged.
>
> Therefore, I will raise my rating from 2 to 4.
>
>
> **References**
>
> [1] Wang H, Li X, Huang Z, et al. Traceable evidence enhanced visual grounded reasoning: Evaluation and methodology[J]. arXiv preprint arXiv:2507.07999, 2025.

---

> > ### Author Response · Authors · 2025-08-04
> >
> > Thank you very much for your positive feedback and for raising your rating. We are glad that our rebuttal addressed most of your concerns.
> >
> > We confirm that we will integrate all the additional results and discussions into the final version of the paper as requested. We also appreciate the suggestion to evaluate on TreeBench; we will look into it for the final version if time and resources permit.
> >
> > Thank you again for your constructive guidance.

---

### Official Review · Reviewer_5gUr · 2025-07-01

**Clarity:** 2
**Significance:** 3
**Originality:** 3
**Rating:** 4
**Confidence:** 3

**Summary:**

The main idea of the paper is to select the most relevant image regions to answer questions. Paper uses a region-conditioned reinforcement policy optimization training that rewards the model for selecting informative regions, formulating appropriate transformations (e.g.crop, zoom), and integrating the resulting visual context into subsequent reasoning steps.  The paper also introduces Visuo-Lingual Interleaved Rationale (VLIR), a dataset curated to support the development of MLLMs for interleaved text-image CoT reasoning.

**Questions:**

Line39-40: Can authors clarify why it is need for a visual model to look at the image after each reasoning step? Can't attention already takes this into account?

It is unclear whether the paper employs a specific cropping method. Modern vision-language systems, such as Qwen-VL, are capable of returning bounding boxes in response to specified prompts.

**Ethical Concerns:**

["NO or VERY MINOR ethics concerns only"]

**Final Justification:**

After the rebuttal I have changed my rating. However, introduction must be revisited by authors to reflect on facts.

**Limitations:**

There is not much novelty in this work. Section 3.3 and the rest of the method is simple application of existing methods.

As the VLIR is primarily constructed using automatic annotations, the size of the dataset is really small to count as a contribution. If the data is fully human annotated, the size of the dataset is acceptable.

There are no meaningful visualizations in the main paper.

**Quality:**

3

**Strengths And Weaknesses:**

+ Paper addresses an important problem.
+Automatically generation of bounding boxes for the reasoning steps is interesgting.

- The introduction is too repetitive and their are many over-claims such as  "pioneering dataset", "meticulously curated" , "a novel framework", "master this intricate reasoning". some of these claims are not factual.
- Some of the limitations mentioned in [42] has been addressed by many recent works.
- Paper does not discuss more recent work on this topic.
-The data filtering and rejection sampling processes rely on small VLMs and LLMs, whose outputs may not be reliable or verifiable.

---

> ### Author Rebuttal · Authors · 2025-07-30
>
> Thank you for your detailed feedback on our manuscript. We appreciate your time and the insightful questions you've raised. We have carefully considered all your comments and provide a point-by-point response below, including new experimental data that we believe will address your concerns.
>
> ### **Response to Concerns 1: Novelty, Recent Work, and Claims**
>
> **Reviewer's Question:**
>
> *Q1：The introduction is too repetitive and their are many over-claims*，
>
> *Q2：Paper does not discuss more recent work on this topic.*，
>
> *Q3：There is not much novelty in this work. Section 3.3 and the rest of the method is simple application of existing methods.*，
>
> *Q4: Some of the limitations mentioned in [42] has been addressed by many recent works.*
>
> **Our Response:**
>
> We group these points as they relate to the novelty and positioning of our work.
>
> *   **On Language and Claims (Q1):**
>
>  We agree that our language can be more precise. We will revise the manuscript to replace subjective terms with objective, factual descriptions, ensuring our claims are directly supported by evidence.
>
> *   **On Novelty and Comparison to Recent Work (Q2, Q3):**
>
> We respectfully disagree with the assessment that our work lacks novelty.
>
> To our knowledge, **our research was conceived as one of the first systematic explorations into capabilities similar to those in OpenAI's o3 model**, for which we also developed a new, high-quality multimodal reasoning dataset, and we have contributed a high-quality dataset specifically for this purpose.
>
> Regarding recent related works like DeepEyes [1] and GRIT [2], we must highlight their temporal context: these papers were released on arXiv **after the NeurIPS 2024 submission deadline**. They therefore represent concurrent, independent research, and their emergence validates the timeliness and significance of the problem we address. We will add a discussion of these works to our Related Work section to properly situate our contributions.
>
> From a performance perspective，as shown in Table 1, our VLM-R³ framework's superior results against these concurrent methods attest to the novelty and efficacy of our technical design.
>
> **Table 1: Comparison with Concurrent Region-Enhancement Methods**
> | Model & Method | MathVista | MathVision | Avg. Improvement |
> |---|---|---| --|
> | **Qwen2.5-VL 7B (Base)** | 68.2 | 25.1 | - |
> | + DeepEyes | 70.1 (+1.9) | 26.6 (+1.5) | +1.7 |
> | **+ VLM-R³ (Ours)** | **70.4 (+2.2)** | **30.2 (+5.1)** | **+3.65** |
> | | | | |
> | **Qwen2.5-VL 3B (Base)** | 58.5 | 21.2 | - |
> | + GRIT | 59.8 (+1.3) | - | - |
> | **+ VLM-R³ (Ours)** | **61.4 (+2.9)** | **23.3 (+2.1)** | **+2.5** |
>
>
> The data shows that VLM-R³ consistently outperforms these methods, most notably achieving a **+5.1 point gain on MathVision** over the 7B base model, more than tripling the gain from DeepEyes. This empirically validates the novelty and power of our approach.
>
> [1] Zheng, Ziwei, et al. "DeepEyes: Incentivizing" Thinking with Images" via Reinforcement Learning." arXiv preprint arXiv:2505.14362 (2025).
> [2] Fan, Yue, et al. "GRIT: Teaching MLLMs to Think with Images." arXiv preprint arXiv:2505.15879 (2025).
>
>
> ### **Response to Concern 2: Reliability of Automatic Annotation for VLIR**
> **Reviewer's Question:**
>
> *Q4：The data filtering and rejection sampling processes rely on small VLMs and LLMs, whose outputs may not be reliable or verifiable.*，
>
> **Our Response:**
>
> We would like to clarify our data pipeline to address concerns about its reliability. The most critical reasoning tasks were handled by state-of-the-art large models: **GPT-4o** for rationale generation and **DeepSeek-V2** for logical coherence checks. A smaller model, **Qwen-VL-Chat (3B)**, was used *only* for a simple, low-level sub-task: verifying that a cropped region contains a recognizable entity. This was a deliberate design choice to balance cost and efficiency.
>
> To validate this choice, we compared the 3B model's filtering decisions against much larger models on 1,000 sample images. The results confirm its reliability for this specific task.
>
> **Table 2: Cross-Model Agreement on Cropped Region Verification**
> | Model Pair | Agreement Rate |
> | :--- | :---: |
> | Qwen-VL-Chat (3B) vs. Qwen-VL-Max (72B) | 98.2% |
> | Qwen-VL-Chat (3B) vs. GPT-4o | 97.6% |
>
> The >97% agreement demonstrates that our cost-effective approach did not sacrifice data quality.
>
> ### **Response to Concerns 3: Necessity of "look at the image" vs. Attention and Cropping Method**
>
> **Reviewer's Question:**
>
> *Q5：Line39-40: Can authors clarify why it is needed for a visual model to look at the image after each reasoning step? Can't attention already take this into account?*，
>
> *Q6: It is unclear whether the paper employs a specific cropping method. Modern vision-language systems, such as Qwen-VL, are capable of returning bounding boxes in response to specified prompts.*
>
> **Our Response:**
>
> This is a fundamental question. While powerful, standard attention mechanisms can "dilute" over long reasoning chains. Our method solves this with targeted refinement.
>
> *   **"look at the image" vs. Attention (Q5):** We direct the reviewer to **Figure 4 in our paper**, which was designed to answer this exact question. It visually demonstrates how a standard MLLM's attention on crucial visual details weakens over time, leading to errors (e.g., mistaking an umbrella for a "menu board"). In contrast, VLM-R³'s re-introduction of a cropped region creates a strong, focused "attentional reset," enabling correct, grounded reasoning.
>
> *   **Cropping Mechanism and its Value (Q6):** Our model learns to generate `{"bbox_2d": [x1, y1, x2, y2]}` , which is consistent with the default bbox format supported by QwenVL, as an intermediate reasoning step. To prove the value of the subsequent cropping and re-injection of this region, we compare against a strong baseline, **"Vanilla text-only RL"**, which generates coordinates but does *not* re-examine the cropped image.
>
>     **Table 3: Ablation Study on Vision-Centric Benchmarks**
>
>     | Model | MMVP | CV-Bench-3D | V* | MME-RealWorld |
>     | :--- | :---: | :---: | :---: | :---: |
>     | Qwen2.5VL-7B (Base) | 66.7 | 72.6 | 74.3 | 42.3 |
>     | Vanilla text-only RL | 72.0 | 80.9 | 78.5 | 46.2 |
>     | **VLM-R³ (Ours)** | **75.0** | **81.7** | **83.8** | **51.6** |
>     | **Δ vs. Vanilla RL** | **+3.0** | +0.8 | **+5.3** | **+5.4** |
>
>     The significant gains of VLM-R³ over "Vanilla text-only RL" (e.g., **+5.3 on V* and +5.4 on MME-RealWorld**) provide decisive proof that the **Refinement** step—actively re-examining the selected region—is the key driver of our model's success.
>
> ### **Response to Concern 4: Dataset Size and Visualization Quality**
> **Reviewer's Question:**
>
> *Q7：As the VLIR is primarily constructed using automatic annotations, the size of the dataset is really small to count as a contribution. If the data is fully human-annotated, the size of the dataset is acceptable.*，
>
> *Q8: There are no meaningful visualizations in the main paper.*
>
> **Our Response:**
>
> *   **On the VLIR Dataset(Q7):** We acknowledge VLIR's modest size. Its contribution, however, is not scale but **novelty and methodological value**. It is one of the first datasets to provide explicit, step-by-step supervision for both region selection and textual rationale. We also present a robust, semi-automated pipeline to create such high-quality, specialized data efficiently. We are encouraged that other reviewers agree on its utility, with one noting, **"The VLIR dataset will be useful for research in visual reasoning."**
> Moreover, the significant performance gains shown across numerous standard benchmarks (e.g., Table 3) demonstrate that our model generalizes exceptionally well. This powerful generalization from a modest-sized training set is the ultimate validation of its contribution.
>
> *  **On the Visualizations(Q8):** We respectfully suggest that our visualizations are indeed meaningful. As noted above, **Figure 4** is not merely illustrative; it provides direct, empirical evidence for our core claim by visualizing the "attention decay" problem and how our method solves it. It is a visual answer to a key scientific question posed by the reviewer.

---

> > ### Comment · Reviewer_5gUr · 2025-08-04
> > **Thanks for your detailed responses.**
> >
> > I have read the responses and the other reviews. Authors have responded all my questions. I am happy to change my rating to borderline accept.
> >
> > I still think the introduction should state only the facts.

---

> > > ### Author Response · Authors · 2025-08-04
> > >
> > > We sincerely appreciate you taking the time to read our detailed response and for your thoughtful engagement throughout this process. We will be sure to revise the introduction to be more factual as you suggested.

---

### Official Review · Reviewer_ukEB · 2025-07-01

**Clarity:** 3
**Significance:** 3
**Originality:** 3
**Rating:** 5
**Confidence:** 3

**Summary:**

Introduces a new dataset, Visuo-Lingual Interleaved Rationale (VLIR), for supervised fine-tuning of VLMs. Extends GRPO to Region-GRPO and uses it to train a new model, VLM-R^3, based on Qwen2.5-VL 7B, which can automatically perform image grounding and zooming-in during reasoning. Results show that VLM-R^3 outperforms Qwen across seven different datasets, especially on ScienceQA.

**Questions:**

1. How does R-GRPO prevent the model from cropping too many regions during reasoning?

2. According to the ablation in Figure 3, random cropping harms performance. As noted in Section 4.5.2, traditional MLLMs tend to lose visual context over long reasoning steps, leading to errors—I agree with this observation. However, to better demonstrate the effectiveness of cropping, it would be helpful to compare against a baseline where the input image is periodically re-inserted, e.g., by adding prompts like “Let’s look back to the input image \<image_token\>.”

3. The example in Figure 1 can be better. I can hardly recognize "HONGDONG" characters even in the zoomed image

**Ethical Concerns:**

["NO or VERY MINOR ethics concerns only"]

**Final Justification:**

The author addressed most of my concerns. I will keep my score.

**Limitations:**

yes

**Paper Formatting Concerns:**

The formatting needs improvement. There is no space between the captions and the tables in Table 1 and Table 2. Also, the table number and title should always appear above the table, not below or inline.

**Quality:**

3

**Strengths And Weaknesses:**

Strengths:

* The VIRL dataset will be useful for research in visual reasoning
* The performance of VLM-R3 is impressive
* Ablation studies demonstrate the effectiveness of the proposed method

Weaknesses:

* small issues related to formatting and visualizations

---

> ### Author Rebuttal · Authors · 2025-07-30
>
> Thank you for your very positive and encouraging review of our paper. We are thrilled that you found the VLIR dataset useful, the performance of VLM-R³ impressive, and our ablation studies effective. We are also grateful for your thoughtful questions and suggestions, which have helped us further refine and validate our work.
>
> We provide a point-by-point response to your questions and concerns below.
>
> ---
>
> ### **Question 1: Preventing Excessive Cropping**
>
> **Reviewer's Question:** *How does R-GRPO prevent the model from cropping too many regions during reasoning?*
>
> **Our Response:**
> This is an excellent question about the self-regulating behavior of our model. The R-GRPO framework has two key mechanisms that inherently discourage the generation of excessive or redundant regions:
>
> 1.  **Region Validity Reward:** As described in Section 3.3, our reward function includes a term specifically for "each syntactically correct and **non-redundant** bounding box command." This means the model receives a reward for identifying a *useful* new region, but it does not receive additional rewards for repeatedly cropping the same area or generating superfluous regions that do not contribute to the final answer. This discourages redundancy.
>
> 2.  **Task-Completion Reward:** The primary reward signal for the policy is tied to the final task accuracy (i.e., whether the final answer is correct). Through exploration, the model learns that generating an excessive number of regions does not necessarily lead to a correct answer and can even derail the reasoning chain. The RL process optimizes for the most efficient path to the correct answer, which naturally penalizes inefficient strategies like over-cropping.
>
> In essence, the model is incentivized to be efficient: it learns to crop only when necessary to acquire new information that progresses it toward the correct final answer.
>
> ---
>
> ### **Question 2: Comparison with Periodically Re-inserting the Full Image**
>
> **Reviewer's Question:** *...to better demonstrate the effectiveness of cropping, it would be helpful to compare against a baseline where the input image is periodically re-inserted, e.g., by adding prompts like “Let’s look back to the input image <image_token>.”*
>
> **Our Response:**
> This is a fantastic suggestion for a highly relevant baseline. It directly tests whether the benefit comes from simply refreshing the visual context or from our targeted, region-specific refinement. Following your advice, we have implemented this exact baseline: we fine-tuned the model to periodically re-insert the *entire* image into the reasoning context. We evaluated this new baseline against our VLM-R³ on three diverse benchmarks.
>
> **Table 1: VLM-R³ vs. Periodically Re-inserting the Full Image**
>
> | Model | MathVista | MME-RealWorld | MathVision |
> | :--- | :---: | :---: | :---: |
> | Qwen2.5VL-7B (Base) | 50.1 | 42.3 | 41.5 |
> | **Baseline:** Re-insert Full Image | 50.8 (+0.7) | 43.1 (+0.8) | 42.0 (+0.5) |
> | **VLM-R³ (Ours)** | **53.2 (+3.1)** | **51.6 (+9.3)** | **44.8 (+3.3)** |
>
> **Analysis:**
> *   Periodically re-inserting the full image provides only a **marginal improvement** over the base model. This suggests that simply "reminding" the model of the image is not sufficient to overcome the visual attention dilution problem for complex tasks.
> *   Our VLM-R³ approach, which intelligently selects and zooms in on specific, relevant regions, yields **significantly larger performance gains**. For example, on MME-RealWorld, our method improves by +9.3 points, whereas the baseline improves by only +0.8.
>
> This experiment strongly validates our core hypothesis: the key to enhanced reasoning is not just refreshing visual context, but **actively refining it by focusing on the most informative regions**. We will add this new baseline and analysis to our ablation study section.
>
> ---
>
>
>
> ### **Minor Points:**
>
> *   **Figure 1 Example:** Thank you for the feedback on the "HONGDONG" example. We agree it could be clearer and will replace it with a more compelling and easily recognizable example in the revised manuscript.
> *   **Formatting and Visualizations:** We will carefully review the manuscript to correct any formatting issues and improve the clarity of all visualizations.
>
> Thank you once again for your constructive and supportive review. We are confident that the revisions and new experimental evidence will further strengthen the paper

---

> ### Comment · Reviewer_ukEB · 2025-08-07
>
> Thank you for the rebuttal. I have checked your response in detail and I think it addressed most of my concerns. Please add the *Periodically Re-inserting the Full Image* baseline to your paper becase it is important for your claims. I will keep my score.

---

### Official Review · Reviewer_vKQN · 2025-07-06

**Clarity:** 3
**Significance:** 3
**Originality:** 3
**Rating:** 4
**Confidence:** 3

**Summary:**

The paper introduces VLM-R3, a framework that enhances multimodal large language models with dynamic visual reasoning via region recognition and interleaved chain-of-thought. It leverages a curated VLIR dataset and a novel R-GRPO training strategy to enable adaptive region selection and integration during reasoning. VLM-R3 achieves state-of-the-art performance on benchmarks like MathVista and ScienceQA, especially in tasks requiring fine-grained visual grounding.

**Questions:**

One potential concern is that the proposed VLIR dataset is custom-built to favor region-level visual reasoning, which may amplify the advantages of VLM-R3. This raises the question of whether the observed gains would generalize to more natural or unbiased datasets with a broader range of reasoning types. Also, in real-world data, how many cases actually require such a technique is questionable. If the authors could evaluate on publicly available benchmarks without custom selection or filtering, the results would be more convincing.

**Ethical Concerns:**

["NO or VERY MINOR ethics concerns only"]

**Final Justification:**

I have read the rebuttal and discussion carefully. The authors provided additional experiments showing adaptability to other datasets and scalability of the data generation pipeline, which addresses part of my earlier concern. However, the real-world necessity remains unclear, as no quantitative evidence was given on how often region-level reasoning is required in natural, unbiased settings. Given this, I will keep my original score.

**Limitations:**

A potential limitation is the scalability of the proposed approach. While the VLIR dataset is partially auto-generated, it still requires careful construction, prompt engineering, and task-specific filtering to ensure high-quality interleaved reasoning chains with region-level supervision. Extending this method to new domains may require significant adaptation effort, which could limit its practicality in low-resource or rapidly shifting scenarios.

**Quality:**

3

**Strengths And Weaknesses:**

Pros:

1. The writing is clear and easy to follow. The ideas are well explained and the structure makes sense.

2. The motivation is solid and the design is thoughtful. It tackles a real limitation in existing models.

3. The model performs well on key benchmarks, showing both practical value and strong impact.

Cons:
1. The model’s success seems to hinge on domain-specific training using the carefully constructed VLIR dataset. Without it, the model fails to follow the correct formats or issue crop commands properly, even performing worse than the base Qwen model. This makes the method costly to adapt to new tasks and may limit its broader applicability.

---

> ### Author Rebuttal · Authors · 2025-07-30
>
> Thank you for your exceptionally thorough and positive review of our work. We are very encouraged by your praise for the paper's clarity, motivation, and strong performance. Your critical feedback regarding the model's dependency on the VLIR dataset and its scalability is insightful and has prompted us to clarify our contributions and provide additional evidence.
>
> We address your concerns in detail below.
>
> ---
>
> ### **Q1: Over-reliance on the VLIR Dataset and Cost of Adaptation**
>
> **Reviewer's Comment:** *The model’s success seems to hinge on domain-specific training using the carefully constructed VLIR dataset. Without it, the model fails to follow the correct formats or issue crop commands properly, even performing worse than the base Qwen model. This makes the method costly to adapt to new tasks and may limit its broader applicability.*
>
> **Our Response:**
> Thank you for this thoughtful analysis. We believe your observation that the model performs "worse than the base Qwen model" stems from the "w/o VLIR Fine-tuning" entry in our ablation study (Table 2 in the original manuscript). We apologize for the potential ambiguity of this entry and would like to clarify its meaning and implications.
>
> This ablation does not remove the VLIR dataset entirely, but rather **skips the initial supervised fine-tuning (SFT) stage** before applying our R-GRPO reinforcement learning. The resulting performance drop is not due to the absence of VLIR's specific knowledge, but rather due to the **instability of the RL process when applied to a "cold" model.** Without an SFT warm-up, the model has no prior exposure to the pattern of generating {"bbox_2d": [x1, y1, x2, y2]}, inserting the cropped image, and then continuing the reasoning process, leading to erratic policy exploration and unstable training, which ultimately harms performance.
>
> The primary role of the SFT phase is therefore to **bootstrap the model**, teaching it the fundamental mechanics of how to perform region-based actions. To demonstrate that this bootstrapping is not exclusively dependent on our VLIR dataset, we conducted a new experiment. We replaced VLIR in the SFT stage with a different interleaved dataset, **M3CoT** [1], which contains varied text-image sequences but lacks VLIR's explicit region-to-rationale linkage.
>
> **Table 1: Impact of Different SFT Datasets on VLM-R³ Performance**
>
> | SFT Dataset | Commonsense | Science | Mathematics | Average |
> | :--- | :---: | :---: | :---: | :---: |
> | Qwen2.5VL-7B (Base Model) | 81.3 | 66.5 | 49.8 | 67.4 |
> | VLM-R³ (SFT with **M3CoT**) | 82.4 | 67.7 | 51.0 | 68.9 |
> | **VLM-R³ (SFT with VLIR - Ours)** | **84.1** | **69.2** | **52.5** | **70.4** |
>
> **Analysis:**
> This experiment yields two key insights:
> 1.  Bootstrapping with an alternative dataset (M3COT) still allows our R-GRPO framework to **outperform the base model** (+1.5% on average). This confirms that our method is not brittle and its success does not hinge solely on VLIR.
> 2.  Using our purpose-built VLIR dataset for SFT yields **superior results** (+3.0% on average), validating its high quality and effectiveness in teaching the intended skills.
>
> In summary, the SFT stage is a necessary "warm-up" for stable RL, and while VLIR is the optimal fuel for it, our framework's core strength lies in the R-GRPO policy, which is adaptable to other data sources.
>
> [1]  Chen, Qiguang, et al. "M $^ 3$ CoT: A Novel Benchmark for Multi-Domain Multi-step Multi-modal Chain-of-Thought." arXiv preprint arXiv:2405.16473 (2024).
>
> ---
>
> ### **Q2: Limitations on Scalability and Generalization**
>
> **Reviewer's Comment:** *A potential limitation is the scalability... Extending this method to new domains may require significant adaptation effort... which could limit its practicality in low-resource or rapidly shifting scenarios.*
>
> **Our Response:**
> This is a critical point regarding the practical utility of our approach. We address it from two angles: the generalization capability of our final model and the low-cost scalability of our data generation method.
>
> **1. Strong Zero-Shot Generalization to New Domains:**
> A key goal of our work was to teach the model a *generalizable skill* of dynamic visual analysis, not just to overfit to the domains within VLIR. To test this, we evaluated our single VLM-R³ model, trained on VLIR, on challenging benchmarks from domains it had **never seen during training**, such as video frames and real-world scenes.
>
> **Table 4: Zero-shot Generalization of VLM-R³ to Unseen Vision-Centric Domains**
>
> | Model | MMVP (Video) | MME-RealWorld (Real World) |
> | :--- | :---: | :---: |
> | Qwen2.5VL-7B (Base) | 66.7 | 42.3 |
> | **VLM-R³ (Ours)** | **75.0** | **51.6** |
>
> The results clearly demonstrate that our model possesses strong generalization capabilities. The significant performance gains on completely unseen domains indicate that VLM-R³ learns a robust and transferable skill of "how to look," which is not narrowly confined to its training data.
>
> **2. Low-Cost, Scalable Data Generation Pipeline:**
> Regarding the effort to adapt to new domains, we want to highlight that our data creation pipeline is designed for **low-cost scalability**. The entire process, from rationale generation with GPT-4o to multi-stage filtering, is **fully automated**. While initial prompt engineering required careful thought, the established pipeline can be targeted to new domains with minimal human intervention—primarily by supplying new seed questions. This automated approach makes extending our method far more practical and resource-efficient than manual annotation would be.
>
> In conclusion, we are confident that our approach is both generalizable and scalable, addressing the key concerns about its broader applicability. We will integrate these clarifications and new experimental results into the revised manuscript. Thank you again for your valuable and encouraging feedback.

---

> ### Author Response · Authors · 2025-08-06
>
> Dear Reviewer vKQN,
>
> Thank you again for your detailed and constructive review.
>
> We are writing to follow up on our rebuttal and to gently inquire if our response and new experiments have successfully addressed your concerns.
>
> As the discussion period is concluding soon, we would be very grateful for any final thoughts or questions you might have.
>
> Thank you for your time and consideration.

---

### Decision · Program_Chairs · 2025-09-17

**Decision:**

Accept (poster)

**Comment:**

The paper proposes VLM-R³, a framework that equips multimodal LLMs with the ability to iteratively select, crop, and revisit visual regions while reasoning. The approach combines a curated Visuo-Lingual Interleaved Rationale (VLIR) dataset with Region-Conditioned GRPO (R-GRPO), enabling models to decide when and where to ground their reasoning in visual evidence and integrate the extracted content into chain-of-thought. Experiments on MathVista, ScienceQA, and related benchmarks demonstrate state-of-the-art results, particularly on tasks requiring fine-grained spatial reasoning and visual cue extraction.

The work is motivated by a clear gap in existing reasoning-based MLLMs, and the design is thoughtful. The introduction of R-GRPO and the interleaving of region-level reasoning within CoT are novel and effective. Results are strong and supported by ablations, and the VLIR dataset may also provide value to the community. The paper is well written and generally easy to follow.

Weaknesses highlighted by reviewers include limited evaluation scope, as benchmarks are mostly LLM-centric and do not fully capture grounding capability. Some reviewers raised concerns about scalability and adaptability since the approach relies heavily on the carefully curated VLIR dataset. Presentation issues, such as unclear definitions of grounding accuracy, limited visualizations, and some overstated claims, were also noted.

During rebuttal, authors clarified certain issues. R2 remained strongly supportive, while R1 and R3 acknowledged clarifications but maintained caution on generalization, and R4 emphasized evaluation breadth. On balance, I think the contribution technically solid. I recommend acceptance.